# Propagation and Dissemination Strategies of Transmissible Spongiform Encephalopathy Agents in Mammalian Cells

**DOI:** 10.3390/ijms23062909

**Published:** 2022-03-08

**Authors:** Stefanie-Elisabeth Heumüller, Annika C. Hornberger, Alina S. Hebestreit, André Hossinger, Ina M. Vorberg

**Affiliations:** 1Laboratory of Prion Cell Biology, German Center for Neurodegenerative Diseases Bonn (DZNE e.V.), Venusberg-Campus 1/99, 53127 Bonn, Germany; stefanie-elisabeth.heumueller@dzne.de (S.-E.H.); annika.hornberger@dzne.de (A.C.H.); alina.hebestreit@dzne.de (A.S.H.); andre.hossinger@dzne.de (A.H.); 2German Center for Neurodegenerative Diseases (DZNE), Rheinische Friedrich-Wilhelms-Universität Bonn, Siegmund-Freud-Str. 25, 53127 Bonn, Germany

**Keywords:** transmissible spongiform encephalopathies, prion, PrP, amyloid, virus

## Abstract

Transmissible spongiform encephalopathies or prion disorders are fatal infectious diseases that cause characteristic spongiform degeneration in the central nervous system. The causative agent, the so-called prion, is an unconventional infectious agent that propagates by converting the host-encoded cellular prion protein PrP into ordered protein aggregates with infectious properties. Prions are devoid of coding nucleic acid and thus rely on the host cell machinery for propagation. While it is now established that, in addition to PrP, other cellular factors or processes determine the susceptibility of cell lines to prion infection, exact factors and cellular processes remain broadly obscure. Still, cellular models have uncovered important aspects of prion propagation and revealed intercellular dissemination strategies shared with other intracellular pathogens. Here, we summarize what we learned about the processes of prion invasion, intracellular replication and subsequent dissemination from ex vivo cell models.

## 1. Introduction

Transmissible spongiform encephalopathies or prion diseases are neurodegenerative diseases that are characterized by the deposition of host-encoded prion proteins as highly stable, beta-sheet rich polymeric aggregates in the central nervous system [1]. Prion diseases affect humans and other mammals and can be of sporadic, genetic or infectious origin. In animals, natural infection usually occurs through contact and/or ingestions of contaminated biological materials [2]. In humans, prion diseases are mainly sporadic, with some genetic cases. Both iatrogenic and zoonotic transmissions have been reported [1]. Prions form by aberrant folding of cellular prion protein PrP into highly ordered protein aggregates with self-templating activities. Once formed, prions convert cellular PrP (PrP^C^) into its aggregated isoform PrP^Sc^, thereby indefinitely propagating PrP misfolding [3,4,5]. A so-far unknown process results in fragmentation of formed PrP^Sc^ aggregates, leading to the generation of smaller seeds that can be transmitted to other cells [6]. A peculiar feature of prions is their existence as strains. When introduced into the same mammalian species, prions can cause disease phenotypes that differ in incubation times, clinical presentation, host organ and cell tropism and neuropathological characteristics such as PrP^Sc^ deposition patterns and spongiform degeneration [7]. Once established in the new host, prion strains are quite stable, causing specific clinical phenotypes and neuropathological changes upon subsequent passage. As prions do not contain a nucleic acid genome, strain information is likely enciphered within the three-dimensional fold of the PrP^Sc^ polymer [8]. Indeed, biophysical and biochemical characterization of PrP^Sc^ molecules associated with different prion strains argue that prion strain information is encrypted in conformational variants of PrP polymers [9]. Cell-free experiments further argue that cofactors both promote or restrict formation of certain prion conformations that are associated with different disease phenotypes [10,11,12].

Prions are cell-associated pathogens that transmit to neighboring cells by different routes. These include the potential release of naked PrP^Sc^ molecules [13], transfer to neighboring cells by direct cell contact [14,15] or secretion of PrP^Sc^ in association with extracellular vesicles by infected cells [16]. However, how exactly is PrP^Sc^ taken up by recipient cells and how do PrP-derived prion strains differ in their cellular propagation and dissemination routes? Here we discuss what we know from cellular models on how PrP-derived prions enter and replicate within their target cells.

## 2. Cell Biology of Cellular PrP

The cellular prion protein PrP^C^ serves as a substrate for PrP^Sc^ formation and is essential for prion propagation. It is anchored to the cell membrane by a glycosyl-phosphatidylinositol moiety (GPI). PrP^C^ is predominately expressed in the central nervous system and, to a lesser extent, in the lymphoreticular system, the female genital tract, intestine, colon and thyroid (reviewed in [17]). Several functions for PrP^C^ have been proposed, including neuronal activity and viability, cell adhesion, cell cycle and immune regulation (reviewed in [18]). Interestingly, PrP^C^ also plays a role in cancer development by regulating cancer stem cell properties and chemoresistance [19]. Recent analyses suggest that PrP^C^ functions through interacting and mediating the posttranslational modification of NCAM1, thereby controlling epithelial-to-mesenchymal transition and related plasticity programs [20].

Following synthesis and glycosylation in the endoplasmic reticulum and Golgi, mature PrP^C^ is present on the cell surface. PrP^C^ is mainly found in rafts, membrane microdomains enriched in phospholipids and cholesterol [21,22]. Some PrP^C^ molecules undergo proteolytic cleavage or membrane shedding (reviewed in [23]). PrP^C^ is also secreted from cells in association with extracellular vesicles, small membrane-bound delivery devices for intercellular communication [24]. PrP^C^ can be internalized from the cell surface by different routes in different cell types [24,25,26,27,28,29]. Following internalization by clathrin-, caveolin- or raft-mediated endocytosis, PrP^C^ is transported through early endosomes and either recycled back to the cell surface [27] or trafficked to late endosomes and subsequently to the lysosome for clearance [30,31]. The route of PrP^C^ uptake is at least partially determined by the sorting receptor VSP10P sortilin, which directly interacts with PrP^C^, moves it out of rafts and mediates its transport to the lysosome [30]. Another factor identified is muskelin which directs PrP^C^ towards the lysosome [32]. Following transport to the lysosome, PrP^C^ is degraded with a half-life of approximately 5 to 24 h. Differences in half-lives likely depend on the amount of PrP^C^ expressed by the cells [33,34].

## 3. Cellular Models for Prion Propagation

The cell biology of prion replication is only incompletely understood. In vivo, neurons and astrocytes represent the major targets for prion replication, with certain cells of the lymphoreticular system also acting as hosts [35]. In vitro, however, only few cell lines of diverse origins are permissive to prion replication (Table 1) [16]. These include brain-derived cell lines of neuronal, astroglial, microglial or Schwann cell origin, but also fibroblasts, epithelial or muscle cells (reviewed in [36]). Not all cell lines of these origins can be infected, so that prion susceptibility must be detected empirically (reviewed in [16]). Once infection is established, prions persistently replicate in susceptible cell lines without overt cytotoxic effects. It is unclear, what—other than PrP—controls prion infection in vitro. The relative expression level of PrP^C^ does not appear to be the limiting factor for successful infection [37,38,39]. Susceptibility is both dependent on the cell line and prion strain, with some cell lines capable of propagating one prion strain but being resistant to another one (reviewed in [36]). Importantly, even in prion-permissive cell lines, the percentage of cells that become persistently infected can substantially differ [38,40]. Infection rates can be so low that PrP^Sc^ is undetectable by Western blot, despite prion infectivity being confirmed by mouse bioassay [40]. In these cases, selection of cell clones with increased susceptibility helps to drastically increase attack rates [28,38,41].

Surprisingly, even isogenic clones derived from the same cell line can be highly susceptible to some prion strains, but refractory to others derived from the same host [39,46,47]. This characteristic of cell clones has been successfully used to discriminate prion strains in vitro [55]. One reason for the differences in prion susceptibility is the genomic instability of cell lines, resulting in clonal cell populations with slightly differing genetic make-ups [74]. Cell clones have also helped to uncover some factors governing susceptibility to prion infection [74,81,82,83]. For example, analysis of N2a cell clones uncovered that a network of genes involved in extracellular matrix homeostasis, including genes for sulfation of glycosaminoglycans, was related to increased susceptibility to certain prions [82]. Genes associated with cell proliferation, protein degradation and heparin binding were detected to influence permissiveness of immortalized sheep microglia to ovine prions [81].

The cell biology of prion replication has been mainly studied in permanent cell lines with few prion strains that can efficiently propagate in vitro. First demonstration of PrP^Sc^ formation upon prion infection was achieved in mouse neuroblastoma cell line N2a exposed to RML/Chandler prions [62]. Subsequently, infections were also performed with 22L, as this strain resulted in high infection rates and could be propagated reliably in cell culture [38,39]. More recently, neuronal and astroglial cultures from wildtype or transgenic mice have also been successfully used for infection studies [41,68,84,85]. Because of the lack of species-specific cell culture systems, researchers focused on ectopic expression of species-specific PrP^C^ in heterologous cell cultures. The rabbit kidney epithelial cell line RK13 proved to be an outstanding cellular model for propagation of several prion strains from diverse species [60]. RK13 cells exhibit only limited to no expression of endogenous PrP. Engineered to overexpress murine [60], sheep [46], elk [70] or bank vole PrP [13], these cells became permissive to infection with prion strains propagated in the respective species. Surprisingly, however, infection of RK13 cells overexpressing human PrP with human-derived prions proved ineffective [86]. Similar heterologous systems were established to propagate bovine spongiform encephalopathy prions [87]. What we learn from these heterologous cell models is: (1) that cellular factors required for prion propagation are not necessarily species specific and (2) that also in heterologous systems strain-specific factors control establishment of persistent infections.

## 4. The Infection Process—The Uptake of Prions

Infection of cells with prions is mostly performed with crude brain homogenate, as this proved to be more efficient than purified PrP^Sc^ [88]. One possible reason for the inefficient infection with purified PrP^Sc^ fibrils is that these tend to stick to the cellular membrane for a long time, thereby delaying uptake [56,89,90]. Indeed, one bottleneck for efficient infection appears to be protein aggregate size, as sonication used to break up amyloid fibrils derived from different proteins promotes cellular uptake [71]. Further, mixing of brain homogenate with cationic lipids increases subsequent infection of permanent and primary cells, likely due to better uptake of infectious inoculum due to positive charge [91,92].

So far, no exclusive receptor has been identified that is required for prion uptake. Several lines of evidence suggest that glycosaminoglycans (GAGs) such as heparan sulfates present on the cell surface and in the endocytic system are required for prion propagation (Figure 1) [93]. However, chemical inhibition of GAG biosynthesis has divergent effects on prion uptake, possibly due to different prion strains used for infection or differences in the purification grade of the inoculum [94]. Other potential receptors include the 37 kDa/67 kDa laminin receptor (LRP/LR) [95] and low-density lipoprotein receptor-related protein 1 (Lrp1) [90]. Uptake is not sufficient for infection and also cells lacking PrP^C^ efficiently internalize external PrP^Sc^ [56,89,90,96]. Genetic and chemical manipulation of endocytosis pathways demonstrated that prions are preferentially taken up by clathrin- and caveolin-independent routes or are able to bypass these routes when blocked [28]. Impairment of one internalization pathway can increase alternative pathways, such as macropinocytosis, that allow efficient PrP^Sc^ internalization [28]. Once internalized, some purified PrP^Sc^ enters the endocytic-recycling pathway that transports cargo and receptors back to the cell surface, but the majority is trafficked to the endo-lysosomal pathway [97]. Studies with purified PrP^Sc^ as inoculum suggest that re-direction of the inoculum and/or newly generated PrP^Sc^ to the endocytic-recycling pathway is important for efficient accumulation of newly formed PrP^Sc^ [97]. Impairment of the route of initial internalization influences the outcome of persistent infections in a strain-dependent manner [28]. In L929 mouse fibroblast cells, impairment of clathrin-mediated endocytosis results in decreased infection with mouse-adapted prion strain RML, while it benefits productive infection with strain 22L. While the reason for the different fates of prions is unknown, such manipulations may shunt invading prions to different endo-lysosomal compartments that may or may not contain factors or conditions that affect PrP^Sc^ formation or clearance in a strain-dependent manner.

## 5. Detection of Productive Infections

Acute PrP^Sc^ formation following prion exposure can be a fast process, with de novo PrP^Sc^ formation being detectable within minutes to hours [46,81,82]. Still, initial PrP^Sc^ formation can also occur with prion strains incapable of establishing persistent infections, arguing that processes or factors downstream of cellular uptake regulate productive PrP^Sc^ formation [98]. Persistent infection requires that PrP^Sc^ formation exceeds clearance and cell division, two processes that reduce the net amount of PrP^Sc^ [34,99]. Mitotically active cells are thus ideally suited to monitor ongoing PrP^Sc^ formation rather than aggregate persistence [100]. However, cell division potentially also prevents propagation of certain prion strains in cellula [53]. This might be especially true for human prions, which so far only propagate in slow proliferating stem-cell derived astrocytes [53] or mixed astroglial cultures derived from transgenic mice [91]. It is possible that kinetics of human prion formation in vitro is slower than cell doubling, so that persistent infection cannot be established.

A problem with determining the time point of established prion infection is that exposures are usually performed with excess PrP^Sc^-containing inoculum (usually 1% *w/v* brain homogenate). Consequently, in mitotically active prion cell models, productive prion infection is monitored several cell passages post infection to dilute remaining inoculum. Weak PrP^Sc^ signals by Western blot are usually apparent at early passage and increase in subsequent passages [46]. The expression of antibody-epitope tagged PrP^C^ helped to discriminate inoculum from newly formed PrP^Sc^ and demonstrated the formation of PrP^Sc^ in two different cell lines within 2-3 passages post exposure to 22L prions [38,101]. Further, a combination of fluorescently labelled PrP^Sc^ for infection and antibodies that primarily bind to PrP^Sc^ rather than PrP^C^ demonstrated increased accumulation of total PrP^Sc^ 72 h post exposure [97]. Thus, productive prion infections in permanent cell lines can be monitored approximately within 6–9 days post infection.

Detection of productive prion infection in primary cells requires extensive rinsing of cells, as inoculum cannot be diluted by cell splitting. A gradual increase in PrP^Sc^ signal following prion exposure is indicative of successful infection. Primary neurons and astrocytes exposed to 22L prions showed increased PrP^Sc^ levels 14-21 days p.i. [96,102]. Similar results were obtained with primary cerebellar granule neurons (CGN) from transgenic mice expressing human PrP^C^ exposed to different human Creutzfeldt–Jakob disease (CJD) strains [52]. De novo production of PrP^Sc^ was first observed in stem-cell derived human astrocytes as soon as 3–8 days post exposure to vCJD or sCJD brain homogenate [53]. In another study, however, mixed glial cultures from transgenic mice expressing human PrP^C^ were exposed to human vCJD or sCJD prions, newly-formed PrP^Sc^ was first detected approximately 120 days post infection [91]. Thus, cell system and/or prion strain strongly influence kinetics of the establishment of prion infections.

## 6. The Site of PrP^Sc^ Formation in Persistently Infected Cells

The exact cellular location of PrP conversion is still ill-defined and might differ depending on the cell type or the prion strain. It can also change from acute to persistent infection [28]. Early experiments with persistently infected cells demonstrated that PrP^Sc^ is derived from PrP^C^ that is first present on the cell surface [103,104]. PrP^Sc^’s self-templating property in mammalian cells is related to its membrane tether [105]. Anchorless PrP^C^ is unable to maintain prion propagation in cell culture [106]. Exchange of the GPI-anchor for other membrane tethers prevents conversion of PrP^C^ to its pathologic isoform [106,107]. In persistently infected cells, PrP^Sc^ formation occurs either directly on the cell surface or along the endocytic pathway following internalization. Interestingly, worm-like structures of PrP^Sc^ are detectable on the cell surface of infected cells [108]. PrP^Sc^ has also been found in basically all compartments of the endo-lysosomal system (for a review, see [109]). Recycling endosomes and/or the multivesicular bodies have been proposed as major sites of conversion [110,111,112]. Eventually, in cellular models, PrP^Sc^ is trafficked to the lysosomes for clearance [104,113]. In N2a cells, PrP^Sc^ has a half-life of less than 2 days [34]. Infection experiments with mixed cultures of cerebellar granule cells and astrocytes confirmed the presence of PrP^Sc^ in the endosomal recycling compartment and lysosomes [96].

For multiplication of prions, growing prion aggregates must somehow be fragmented to produce seeds that can be transmitted to daughter cells or bystanders. PrP’s unique location on the cell surface and within endo-lysosomal compartments could enable interaction with cellular factors mediating efficient fragmentation and thereby replication of protein aggregates [6,105]. While such factors have so far not been identified in mammalian cells, it is important to note that the disaggregase Hsp104, which turns protein aggregates in lower eukaryotes into self-templating entities, lacks a homologue in mammalian cells [114]. Any potential fragmentation process must thus be accomplished by other cellular processes.

## 7. Intercellular Dissemination of Prions

Prion strains exhibit selected brain region- and cell-tropism, with some strains preferentially targeting neurons, while others also accumulate in astrocytes [115]. The exact mechanisms of prion spreading in vivo remain elusive and most of our understanding of such processes stems from observations made with cellular models. Interestingly, prion maintenance in cell culture is mainly due to segregation of prions to both daughter cells during cell division [99]. PrP^Sc^ signal intensities on Western blots and percentages of infected cells can increase over multiple cell divisions, demonstrating that prions also horizontally transmit to naïve bystander cells [101].

PrP^Sc^ can be directly transmitted to acceptor cell membranes in close proximity to the infected cell [14]. The association with the cell membrane facilitates spreading of PrP^Sc^ from cell to cell by tunneling nanotubes, thin, transient actin-rich tubes connecting cells for transfer of organelles and endocytic vesicles (Figure 2) [116]. Experiments with 22L-infected primary astrocytes demonstrated that efficient intercellular transfer of PrP^Sc^ to recipient CAD cells was predominately due to close cell contact, suggesting that tunneling nanotubes or other cellular contacts facilitate prion transfer [96]. Direct evidence for this intercellular transfer came from co-culture experiments with a cell line persistently propagating mouse-adapted prion strain 139A [15]. PrP^Sc^ molecules could theoretically traverse intercellular bridges such as tunneling nanotubes to uninfected cells by propagating along the surface of the tubular conduits [108]. However, PrP^Sc^ also colocalizes with endocytic compartments in tunneling nanotubes, suggesting that prions hijack these vesicles for intercellular transmission [117].

Secretion of PrP^Sc^ and/or prion infectivity into the cell culture supernatant has been reported for several (Table 2) but not all cell models [96]. The first demonstration that this infectivity is associated with extracellular vesicles (EVs) came from experiments with two different transgenic cell lines replicating sheep prions [118]. EVs are nano-sized vesicles that are secreted by virtually all cell types. EVs serve as communication devices that transfer different RNA types, lipids and proteins to distant acceptor cells [43]. EVs associated with prion infectivity exhibited the size and density of exosomes, vesicles derived by invagination of endosomal structures termed multivesicular bodies [118]. At least in RK13 cells, 90% of prion infectivity in conditioned medium could be recovered by 100.000× *g* ultracentrifugation, which sediments small EVs with densities corresponding to exosomes, but infectivity was also present in fractions containing larger vesicles and even in the non-pelletable fraction [13]. Prion infectivity is also associated with the 100,000× *g* fraction of conditioned medium from 22L-infected L929 cells (Figure 2). PrP^Sc^ was also found associated with large EVs in another study, suggesting they were expelled from the cell surface [44]. Chemical and genetic manipulation of EV biogenesis in prion-infected cells also affects secretion of infectivity and subsequent infection of target cells [75,119,120]. Interestingly, in RK13 cells overexpressing ovine, mouse or vole PrP and infected with different prion strains, infectivity levels in EV fractions differed markedly [13]. Such differences in prion release could be due to general expression levels of PrP^C^, cell clone differences or, more intriguingly, to differences in the sorting of prion strains through the endosomal pathway. As EVs preferentially bind to and exert their biological function in specific target cells [65,121], it is quite possible that also the target cell tropism of EVs influences intercellular prion spreading. Further experiments with susceptible cells expressing wildtype levels of PrP^C^ and infected with prions from the same species as well as different recipient cell lines will help to clarify these issues.

Once released from their donor cells, EVs transmit cargo following interactions with specific receptors on their target cells. EVs can either directly fuse with the cell membrane, or they are taken up by endo- or macropinocytosis (reviewed in [73]). While EV cell tropism has been reported, only few receptor-ligand interactions mediating cell targeting have been identified. For example, uptake of some EVs is linked to integrin internalization [123]. Importantly, GAGs such as heparan sulfate proteoglycans, essential for prion propagation in cell culture [93,94], also mediate uptake of EVs [124]. The roles of proposed prion receptors Lrp1 or the 37 kDa/67 kDa laminin receptor (LRP/LR) for prion loaded-EVs are unknown. As PrP^Sc^ is exposed on the surface of the EV, the association of the prion-loaded EVs with cell-surface or endosomal PrP^C^ does not require escape from the endo-lysosomal system for initiation of PrP^Sc^ formation. Exact cellular mechanisms of prion infection following EV uptake remain, however, unexplored.

## 8. Role of Viruses in Intercellular Prion Spreading

Growing evidence suggests that viruses or viral proteins have an impact on prion propagation.

A seminal study in mouse fibroblast cells demonstrated that secretion of prion infectivity was strongly enhanced when prion-infected fibroblast cultures were concomitantly infected with murine leukemia virus MuLV [22]. PrP^Sc^ and PrP^C^ both co-localized with retroviral proteins Env and Gag at the cell membrane and were secreted in association with both retroviral particles and EVs. Viral infection strongly increased the release of PrP^Sc^ and infectivity. The strong increase in PrP^Sc^ secretion was attributed to the expression of the viral precursor protein Gag known to drive viral particle formation and enhance EV release. An increase in prion maintenance following retroviral Gag expression was also observed in an RK13 cell model propagating chronic wasting disease prions [51]. A possible explanation for the effect of retroviral Gag on prion secretion is that Gag proteins associated with prion-containing EVs promote their secretion, thereby increasing horizontal prion dissemination. However, downregulation of Gag expression in N2a cells did not affect the release of prion infectivity, arguing that prions were secreted independent of Gag [75]. We recently demonstrated that also viral glycoproteins can drastically increase intercellular transmission of prions and other protein aggregates [72]. Viral glycoproteins such as VSV-G of vesicular stomatitis virus or the spike S protein of SARS CoV-2 mediate receptor-specific target cell binding and subsequent merging of cell membranes or EVs with the cell surface or endosomes of recipient cells. Both VSV-G or spike S associated with the cell surface and EVs and enhanced protein aggregate transfer to recipient cells. Interestingly, viral glycoprotein VSV-G, but not spike S, also increased release of EVs. When prion-infected N2a cells were transfected with VSV-G plasmid, EVs from their conditioned medium strongly increased infection of L929 and CAD recipient cells. This was also the case when EVs from mock- or VSV-G-transfected donors were adjusted for comparable particle numbers. These results demonstrate that both elevated EV numbers and increased EV binding/fusion with the target cell membranes contribute to intercellular prion spreading.

That viruses might play a role in prion dissemination is also supported by findings in vivo. Small retroviruses have been implicated as cofactors that enhance the spread of scrapie by milk to suckling lambs through simultaneous infection of mammary glands with scrapie prions [63]. However, attempts to demonstrate the effect of MuLV retrovirus on prion propagation in mice failed, likely because target cells for virus and prions differ [64,66]. Surprisingly, a recent in vitro study demonstrated a very different mechanism of how viral infections could affect prion biogenesis. In a small percentage of surviving N2a cells infected with influenza virus, spontaneous PrP^Sc^ formation was observed that was maintained upon continuous cell passage [125]. Mice injected with cell lysates succumbed to disease and exhibited full-blown prion pathogenesis. These experiments indicate that viral infections could in fact even trigger initial events leading to spontaneous formation of infectious prions. An important question to answer here is if spontaneous formation of prions is also observed within human cells expressing human PrP^C^.

## 9. Conclusions

Concerted efforts in the last couple of years have provided us with long-awaited cellular models for propagation of bovine and human prions. However, despite important progress made in prion cell models, many unresolved questions remain, for example: we still do not understand which exact factors determine cellular prion permissiveness and why there are strain-specific differences in susceptibility even when the prion strains come from the same host. What is the exact intracellular site of PrP^Sc^ formation? Additionally, what is the link between viral infection, prion biogenesis and dissemination? Results from cell culture models are exciting, as they demonstrate the strong effect of viral proteins on prion spreading.

## Figures and Tables

**Figure 1 ijms-23-02909-f001:**
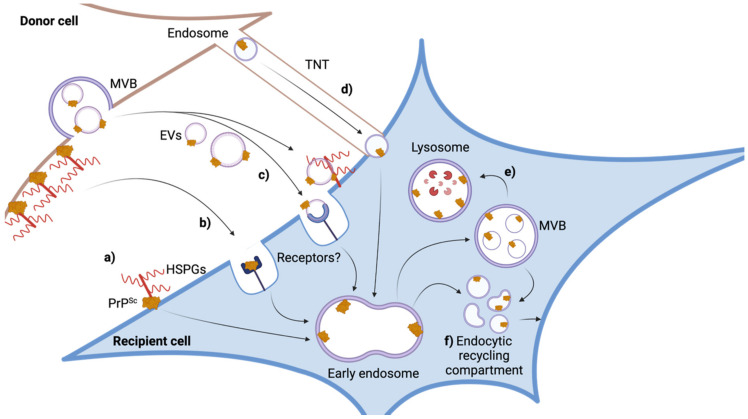
Prion infection mechanisms. Dissemination of PrP^Sc^ relies on different routes. PrP^Sc^ can be transmitted from a donor (brown) to a recipient (blue) cell. Most studies on prion uptake and infection have been performed with purified PrP^Sc^ or with crude brain homogenate containing prions. If PrP^Sc^ is “freely” released into the extracellular space is unknown. (**a**) Receptors for exogenously added PrP^Sc^ include heparan sulfate proteoglycans (HSPGs), Lrp1 or the 37 kDa/67 kDa laminin receptor (LRP/LR). (**b**) “Free” PrP^Sc^ can be internalized by different endocytosis routes or macropinocytosis. (**c**) In cellular systems, PrP^Sc^ can be released from donor cells via microvesicles shedding from the cell surface or in association with smaller extracellular vesicles (EVs) derived from multivesicular bodies (MVBs) that fuse with the cell membrane. EVs can be taken up by recipient cells by different pathways. Few EV ligands that mediate binding to target cells have been identified. Viral ligands present on PrP^Sc^-containing EV can bind to recipient cells and facilitate subsequent infection. (**d**) PrP^Sc^ can also transmit to recipient cells within endosomal vesicles through tunneling nanotubes (TNTs). (**e**) Within target cells, the majority of internalized PrP^Sc^ is directed to the lysosome for degradation. (**f**) Newly formed PrP^Sc^ can be found on the cell surface, within the endocytic recycling pathway and the endo-lysosomal pathway. Productive infection requires PrP^C^ expression but is determined by additional cellular factors and the prion strain. Created with BioRender.

**Figure 2 ijms-23-02909-f002:**
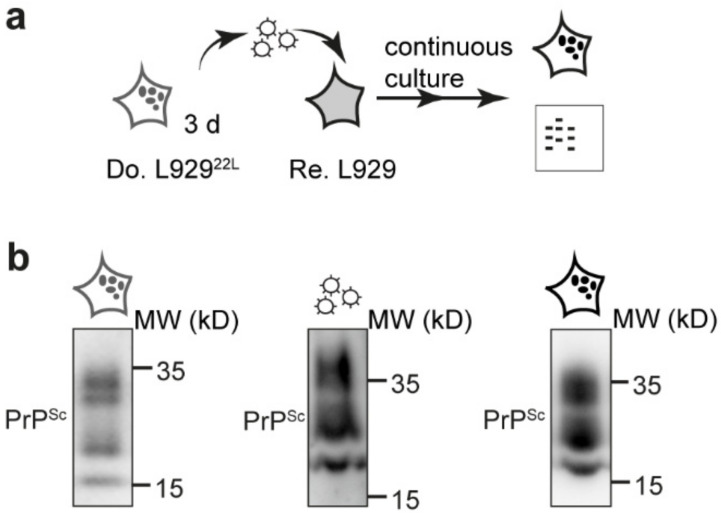
L929 cells infected with 22L secrete PrP^Sc^ and prion infectivity in association with EVs. (**a**) EVs isolated from conditioned medium of L929 cells persistently infected with scrapie strain 22L (L929^22L^) were added to L929 cells. Recipient cells were passaged at least four times before PrP^Sc^ formation was monitored by Western blot. (**b**) Western blot detection of PrP^Sc^ in donor cells (L929^22L^) or PrP^Sc^ in EVs derived from donor cells or PrP^Sc^ in recipient cells after continuous culture. PrP^Sc^ present in proteinase K-treated lysates was detected using anti-PrP antibody 4H1. PrP^Sc^ runs as unglycosylated, monoglycosylated and diglycosylated bands.

**Table 1 ijms-23-02909-t001:** Cell lines susceptible to prions.

Species Inoculum	TSE	Strain	Prion Susceptible Cell Line	Expressed PrP	References
Sheep	Scrapie	Natural Scrapie	RK13, hTERT microglia, MovS6/ MovS2 cells	Ovine	[42,43,44]
			Primary neurons: cerebellar granular, iPSC-derived astrocytes	Ovine	[45]
		127S	RK13, MovS6/ MovS2 cells	Ovine	[13,41]
		PG127	Rov cells	Ovine	[46]
		LA404	Rov cells	Ovine	[46]
		Kanagawa Scrapie	GT1	Murine	[47]
		Obihiro Scrapie	MG20	Murine	[48]
Elk/Deer	Cervid chronic wasting disease (CWD)	Mule Deer CWD (MD-CWD)	CAD5	Cervid	[49]
			MEF	bank Vole/cervid	[49]
			MDB	Mule Deer	[50]
		White-Tailed Deer CWD (WT-CWD)	CAD5	Cervid	[49]
			MEF	Bank Vole/cervid	
		Elk CWD	RK13	Elk	[51]
Cattle	Bovine spongiform encephalopathy (BSE)	BSE	MG20	Murine	[48]
Human	Creutzfeldt–Jakob disease (CJD)	Sporadic CJD	Primary neurons: cerebellar granular, iPSC-derived astrocytes	Human	[52,53]
		Variant CJD	Primary neurons: cerebellar granular, iPSC-derived astrocytes	Human	[52,53]
		Iatrogenic (iCJD)	Primary neurons: cerebellar granular	Human	[52]
Mouse-adapted	Scrapie	Ch./RML	N2a (and subclones), SMB, GT1, CAD5, SN56, 1C11, MG20, C8D1A, MSC-80, L929, RK13	Murine	[38,42,48,54,55,56,57,58,59,60,61,62]
			Primary neurons: cortical, hippocampal	Murine	[56,63,64]
		79 A	N2a PK1 subclone, SMB, CAD5, L929	Murine	[39,40,48,65]
		139 A	N2a (and subclone), SMB, GT1, CAD5, CRBL, L929	Murine	[39,40,48,65,66]
			PC12	Rat	[67]
			Primary neurons: cortical, striatal	Murine	[68]
			Primary neurons: cerebellar granular, astrocytes	Murine/ovine	[41,69]
		22L	N2a (and subclones), GT1, CAD5, SN56, 1C11, HpL3-4, CF10, C8D1A, L929, NIH/3T3, RK13	Murine	[39,40,57,62,70,71,72,73]
			Primary neurons: cerebellar granular, cortical, striatal, hippocampal	Murine	[56,63,74]
		ME 7	N2a subclones, CAD5, SN56, MG20, L929	Murine	[39,48,61,75]
			Primary neurons: cerebellar granular	Murine	[76]
			PC12	Rat	[67]
		22F	SMB	Mouse	[54]
	BSE	301 C	CAD5	Murine	[55]
	CJD	M1000	RK13	Murine	[60,77]
		SY	GT1	Murine	[78]
		FU	N2a, GT1	Murine	[79]
	GSS	Fukuoka 1 (Fu-1)	GT1, 1C11, RK13	Murine	[57,72]
			Primary neurons: cerebellar granular	Murine	[68,76]
Bank vole-adapted	BSE	Bank vole-adapted BSE	RK13	Bank vole	[60]
Hamster-adapted	Transm. mink encephalopathy (TME)	Hyper (HY)	CAD5	Hamster	[80]
	Scrapie	263K	CAD5	Hamster	[80]
		139H	CAD5	Hamster	[80]

Abbreviations: CAD5—mouse catecholaminergic neurons; CF10—mouse neuronal cells; CRBL—mouse cerebellum cells; C8D1A—mouse astrocytic cells; GT-1—mouse hypothalamic neurons; HpL3-4—mouse hippocampal cells; hTERT—immortalized ovine microglia; L929—mouse fibroblasts; MDB—mule deer meningeal fibroblasts; MEF mouse embryonic fibroblasts; MG20 mouse microglia cells; MovS6/MovS2 mouse Schwann cells; MSC-80 mouse Schwann cells; N2a—mouse neuroblastoma cells; PC12—rat pheochromocytoma; NIH/3T3 mouse fibroblasts; Rov—rabbit kidney epithelial cells expressing ovine PrP; RK13—rabbit kidney epithelial; SMB—mouse brain cells, SN56—mouse septal neurons; 1C11—mouse embryonal carcinoma cells (neuronal stem cells).

**Table 2 ijms-23-02909-t002:** Cell lines secreting prion infectivity.

Cell Line	Origin	Prion Strain	EV Isolation Method	PrP^Sc^ Association with EV	EM Confirmation EV	Recipient Cell	Detection of PrP^Sc^ p.i.	Reference
GT1	Mouse hypothalamic neurons	RML	Not isolated, conditioned medium used	n.d.	No	N2a GT1	6–8 weeks	[57]
GT1-7	Mouse hypothalamic neurons	M1000	Differential centrifugation	WB	No	RK13 expressing murine PrP	6 passages	[122]
		M1000	Differential centrifugation	WB	Yes	GT1-7; RK13 expressing murine PrP	One month	[77]
NIH/3T3	Mouse fibroblast+/− infectionMuLV	22L	Differential centrifugation	WB	Yes	NIH/3T3	16 passages Infection only when donors were MuLV infected	[22]
RK13: Rov	Rabbit kidney epithelial cells ectopically expressing ovine PrP	Sheep scrapie PG127	Differential centrifugation	WB	Yes	Rov	Several weeks	[118]
		Sheep scrapie PG127	Differential centrifugation	WB	No	Rov	4 weeks	[13]
RK13: moRK13	Rabbit kidney epithelial cells ectopically expressing murine PrP	22L	Differential centrifugation	WB	No	moRK13	4 weeks	[13]
		M1000	Differential centrifugation	WB	Yes	GT1-7; RK13 expressing murine PrP	One month	[77]
N2a	Mouse neuroblastoma expressing VSV-G	22L	Differential centrifugation	n.d.	No	L929 CAD	7–8 passages	[72]
	Mouse neuroblastoma overexpressing murine PrP	22L	Differential centrifugation	WB	Yes	N2a	3 weeks	[75]
Mov	Immortalized Schwann cell-like cells from transgenic mouse expressing ovine PrP	Sheep scrapie PG127	Differential centrifugation	WB	Yes	Mov	Several weeks	[118]
Hpl3-4 moPrP-3F4	Mouse hippocampus- derived, ectopically expressing epitope-tagged mouse PrP	22L	Not isolated, conditioned medium used	n.d.	No	Hpl3-4 moPrP-3F4	14–28 passages	[101]
L929	Mouse fibroblasts	22L	Differential centrifugation	WB	No	L929	Several weeks	This study

Abbreviations: n.d.—not done; WB—Western blot; p.i.—post infection.

## Data Availability

Data is contained within the article.

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
