# Peer review of "Propagation and Dissemination Strategies of Transmissible Spongiform Encephalopathy Agents in Mammalian Cells"

_ijms, 2022, doi:10.3390/ijms23062909_

Round 1
Reviewer 1 Report
Manuscript: “Propagation and dissemination strategies of transmissible spongiform encephalopathy agents in mammalian cells”.
It is a very interesting concept for propagation and dissemination strategies of transmissible spongiform encephalopathy agents in mammalian cells. I would like to recommend for points require modification and more precise description.
- The manuscript is interesting, but it lacks clear flow with respect to its topic.
- Abstract is not well written not as per title
Overall manuscript needs to be improved.
Author Response
- The manuscript is interesting, but it lacks clear flow with respect to its topic.
Response: Thank you for your review. We believe this must be a misunderstanding due to our abstract and introduction, for which we would like to apologize. In both abstract and introduction, we emphasized too much other neurodegenerative diseases, which might have given the impression we also covered mechanisms of spreading of other proteopathic seeds. We have now drastically shortened those sections to clarify we are only discussing aspects of TSE spreading in cell culture. We have also carefully revised the text to make it more readable. Further, we have changed the order of some subsections for clarity. We believe that we are adequately and comprehensively covering individual steps of prion propagation and dissemination in cellular models.
- Abstract is not well written not as per title
Response: We have now revised the abstract to better summarize the topic of this review and deleted parts on other neurodegenerative diseases which we are not covering in this review.
Overall manuscript needs to be improved.
Response: We have carefully revised the manuscript for clarity.
Reviewer 2 Report
The authors summarized recent works about the processes of prion invasion, intracellular replication and subsequent PrPSc distribution from several cell models with various prion strains. Overall, the description of previous works is clear. Some revision is required before publication.
- This is a review paper. Tables and Figure 1 are very well organized. Figure 2 shows result of Western blotting. This blotting is not shown in reference 71. What is the source of this result?
- As mentioned in page 6, “susceptibility is both dependent on the cell line and prion strain……”. The authors mentioned lots of works on N2a cell and 22 L strain. Why did authors prefer these works?
- Minor change: the format of references and some punctuation (main text and figure 2 legend).
Author Response
The authors summarized recent works about the processes of prion invasion, intracellular replication and subsequent PrPSc distribution from several cell models with various prion strains. Overall, the description of previous works is clear. Some revision is required before publication.
- This is a review paper. Tables and Figure 1 are very well organized. Figure 2 shows result of Western blotting. This blotting is not shown in reference 71. What is the source of this result?
Response: Thank you for your review. We apologize for not mentioning this in the text. Results from figure 2 have not been published before. We have now included a sentence on this data in the text: “Prion infectivity is also associated with the 100,000 x g fraction of conditioned medium from 22L- infected L929 cells (Fig. 2).”
- As mentioned in page 6, “susceptibility is both dependent on the cell line and prion strain……”. The authors mentioned lots of works on N2a cell and 22 L strain. Why did authors prefer these works?
Response: In this specific sentence, we reference a review article that nicely summarizes findings on restricted prion susceptibility of different kinds of cell lines. We have now included a “reviewed in” to cite the review article. We discuss N2a cells in more detail for historical reasons, as many observations were initially made in these cells.
We had included several references to 22L prions, as we and others have found that this strain replicates really well in cell culture, now mentioned in the text. Further, we have now included a sentence describing that N2a cells were the first cell line in which PrPSc could be detected: “First demonstration of PrPSc formation upon prion infection was achieved in mouse neuroblastoma cell line N2a exposed to RML/Chandler prions [50]. Subsequently, infections were also performed with 22L, as this strain resulted in high infection rates and could be propagated reliably in cell culture [40,41].”
Of note, we also extensively cover work in RK13 cells, as well as fibroblasts and primary cells.
Minor change: the format of references and some punctuation (main text and figure 2 legend).
Response: Thank you for pointing this out. We have tried to change punctuation where we find it appropriate. My apologies for wrong punctuation. For the reference format, we have downloaded the endnote reference style MDPI as suggested by the journal. We have now changed the font to Times New Roman.

Round 2
Reviewer 1 Report
The authors have improved the manuscript still, I have the following concerns:
- Introduction: what is rational for starting with NDs, which deviates the reader from the context of the title.
- In the Introduction, the authors need to mention the relation between NDs and encephalopathy.

Author Response
Dear Reviewer
thank you for your review.
The rationale for starting with neurodegenerative diseases is that prion diseases are a subgroup of neurodegenerative diseases (pubmed search for these two terms as of today: 3, 062 hits) that are infectious.
However, to clarify, we have now changed the first sentence of the introduction and state that TSEs are neurodegenerative diseases: "Transmissible spongiform encephalopathies or prion diseases are neurodegenerative diseases that are characterized by the deposition of host-encoded prion proteins as highly stable, beta-sheet rich polymeric aggregates in the central nervous system [1]." and cite a prion review article.
According to NIH, "encephalopathy" is a term for any diffuse disease of the brain that alters brain function or structure. https://www.ninds.nih.gov/Disorders/All-Disorders/Encephalopathy-Information-Page. This term is common in the ND field. As such, we prefer not to define this term further.
